# Feasibility of Smartphone-Based Exercise Training Integrated with Functional Electrical Stimulation After Stroke (SETS): A Preliminary Study

**DOI:** 10.3390/s25041254

**Published:** 2025-02-19

**Authors:** Rudri Purohit, Juan Pablo Appelgren-Gonzalez, Gonzalo Varas-Diaz, Shuaijie Wang, Matias Hosiasson, Felipe Covarrubias-Escudero, Tanvi Bhatt

**Affiliations:** 1Department of Physical Therapy, University of Illinois at Chicago, Chicago, IL 60607, USA; rpuroh2@uic.edu (R.P.); wangshj@gmail.com (S.W.); 2PhD Program in Rehabilitation Sciences and Neuroscience, University of Illinois at Chicago, Chicago, IL 60607, USA; 3Translational Research Unit, Trainfes Center, Santiago 8760903, Chile; jpgonzalez@trainfes.com (J.P.A.-G.); matias@trainfes.com (M.H.); felipe@trainfes.com (F.C.-E.); 4Biomedical Imaging Center, Pontifical Catholic University, Santiago 7820436, Chile; 5Carrera de Kinesiología, Facultad de Medicina, Pontificia Universidad Católica de Chile, Santiago 7820436, Chile; gvaras2@uic.edu; 6Departamento de Kinesiología, Facultad de Arte y Educación Física, Universidad Metropolitana Ciencias de la Educación, Santiago 7760197, Chile

**Keywords:** older adults, stroke, home, rehabilitation, digital technology, walking, strength, balance

## Abstract

One emerging method in home stroke rehabilitation is digital technology. However, existing approaches typically target one domain (e.g., upper limb). Moreover, existing interventions do not cater to older adults with stroke (OAwS), especially those with high motor impairment, who require adjunct therapeutic agents to independently perform challenging exercises. We examined the feasibility of Smartphone-based Exercise Training after Stroke (SETS) with Functional Electrical Stimulation (FES). A total of 12 participants (67 ± 5 years) with stroke (onset > 6 months) exhibiting moderate-to-high motor impairment (Chedoke McMaster Leg ≤ 4/7) underwent 6 weeks of multicomponent (gait, functional strength, dynamic balance) training integrated with FES to paretic lower limb muscles. Primary measures included safety and adherence. Secondary measures included motivation, acceptability and attitude, usability, and clinical measures of gait and balance function like the 10-Meter Walk Test and Mini-BESTest. Participants reported no adverse events and moderate-to-high adherence (84.17 ± 11.24%) and improvement (up to 40%) in motivation, acceptability, and attitude and system usability. Participants also showed pre-post improvements in all measures of gait and balance function (*p* < 0.05). Integrating SETS and FES is feasible and yields short-term gains in gait and balance function among OAwS. Future studies could validate our findings by examining its efficacy with control groups to identify the differential effects of SETS and FES.

## 1. Introduction

Stroke is one of the leading causes of long-term disability, thus, remains a public health concern [1,2,3,4]. Stroke incidence and severity increases exponentially with age, with over 50% of stroke occurrences in people over 75 years of age [5,6]. Motor impairments that limit movement on one side of the body (hemiparesis) affect up to 80% of older adults with stroke (OAwS) [7,8] and are related to gait and balance dysfunction [9,10,11]. Gait and balance dysfunction may further limit functional mobility [12,13], reduce physical activity [14,15], and restrict community participation [16,17]. Exercise-based rehabilitation interventions (e.g., physical therapy—PT) are a promising approach to induce reductions in gait and balance dysfunction post-stroke [18,19,20]. However, over 50% of OAwS exhibit ongoing gait and balance dysfunction after discharge from outpatient PT [7,21]. Further, only few OAwS continue to engage in exercise-based interventions after outpatient discharge [21], possibly due to some barriers. Specifically, these include access barriers to clinic-based programs or community-based programs such as stroke support groups, healthcare barriers such as lack of insurance coverage due for performance plateau, lack of programs in low-income areas, and/or personal barriers such as the inability to independently travel or afford transportation, reliance on a caregiver. A discontinuity in exercise participation post outpatient PT discharge could have negative consequences on physical and psychosocial health such as limited functional mobility, reduced community participation, and decreased quality of life [22,23,24]. Hence, providing extended interventions after outpatient PT discharge, especially those that can be delivered at home, might assist in reducing gait and balance dysfunction among OAwS.

Cumulative evidence from multiple systematic reviews and meta-analysis have shown benefits of exercise-based home interventions (both supervised and unsupervised) on upper limb function (*n* = 26 studies) [25], physical activity (n = 18 studies) [26], and activities of daily living (*n* = 49 studies) [27] among people with stroke. In addition, there is some evidence on home interventions (*n* = 12) targeting gait and balance dysfunction among adults with stroke [28]. Current home-based interventions are mainly supervised, either delivered by healthcare professionals (e.g., physical or occupation therapists, wellness coaches) [29,30,31,32] or by caregivers [33,34]. Certain limitations exist for both current supervised and unsupervised home-based interventions among people with chronic stroke. Specifically, supervised home-based interventions might be prone to scheduling restrictions for the participant [31,32,33,34] and are less likely to promote autonomy to continue exercises after termination possibly due to high dependence on interventions. Unsupervised home-based interventions can overcome the above stated barriers of supervised interventions; however, they typically demonstrate low adherence and involve components that mainly target upper limb function and general health. An integrative review showed that only about 25% of participants receiving unsupervised home-based interventions demonstrated high adherence and corresponding improvements in outcomes of upper limb function (e.g., Wolf Motor function) or general health (e.g., Barthel Index) [35]. These limitations could be due to low participant motivation, accountability, and less engaging intervention delivery [30,36,37,38,39]. One emerging solution to overcome the barriers of unsupervised home-based interventions is the implementation of digital technology.

Over the past five years, digital technology has emerged as an effective tool to deliver unsupervised home interventions among people with stroke. Smart application-based interventions are to be more effective for improving physical activity (number of steps) and motor function than conventional clinic-based PT [40,41,42,43,44]. Further, smart application-based interventions have shown lower dropout rates than other unsupervised home exercise programs that were delivered via paper booklets or brochures [45]. A recent 2023 systematic review concluded that digital mobile apps led to significantly greater improvements in physical function in adults with stroke than conventional clinic-based PT [46]. However, out of the 11 studies, only 7 were conducted in adults with chronic stroke, out of which 4 were delivered in unsupervised, home-based settings. Most existing interventions leveraging smartphone applications have focused on promoting physical activity (e.g., number of steps per day) or improving upper extremity function [41,43,45] rather than improving functional mobility, gait, or balance control. Only two studies incorporated smartapp technology into home-based balance training, which involved sitting or standing postural sway exercises [42,47]. Digital technological interventions have mainly enrolled a majority of younger stroke populations (<65 years) and those with high functional mobility (e.g., motor impairment on CMSA > 5/7) [48,49,50]. Very few studies have examined digital home interventions specifically among marginalized populations including OAwS and those with high motor impairment. OAwS with high motor impairments are likely to be functionally less mobile but might be most in need of exercise interventions that can be safely practiced in home-based settings [51,52,53]. Some of the barriers to including OAwS in high stroke-related motor impairment might stem from safety concerns, inability to perform challenging functional exercises independently, and lack of ease of intervention implementation [54,55].

One way to overcome these barriers among OAwS with high motor impairment could be the use of adjunct therapies such as robotic devices, mirror therapy, or electrical stimulation [56,57,58]. For example, Functional Electrical Stimulation (FES) delivered synchronously with functional tasks is a non-invasive and widely used adjunct agent to enhance exercise performance post-stroke. In the past 5 years, over 10 randomized controlled trials have examined the effect of FES with exercise training and showed greater improvements in lower limb motor function, reductions in muscle spasticity, and improvements in isometric muscle strength than training without FES [59,60,61,62,63,64]. However, most of the previous interventions have been clinic-based and used wired-FES systems [59,60,61,62,63,64] and might have restricted ability to translate to the home setting. To our knowledge, only four studies have implemented FES in the home setting among individuals with stroke [65,66,67], most of which focused on upper limb function. Hence, it remains unknown whether FES-integrated exercise training can be delivered in home settings to target gait and balance dysfunctions among OAwS with high motor impairments.

With recent advances in technology, FES with wireless sensors have been developed to trigger stimulation in synchrony with diverse functional tasks. FES integrated with digital technology now has the potential to induce immediate effects on outcomes of gait and balance control among OAwS with high motor impairments. The application of FES to paretic muscles can immediately stimulate the peripheral neuromuscular units via activation of muscle spindles and neuromuscular junctions, which can assist with the completion of challenging gait and functional balance tasks that might otherwise be difficult to complete for OAwS with high motor impairment [68,69,70]. The neurofacilitatory effect of FES can potentially assist in promoting adherence to the intervention by enhancing voluntary movement control and improving performance on challenging functional tasks potentially via its peripheral and central mechanisms. When applied to weakened paretic muscles, FES activates neuromuscular units (e.g., muscle spindles, neuromuscular junctions) [71], proprioceptive pathways, which could produce functional movements otherwise not possible without external assistance, and provide sensory feedback that improves motor control and coordination. Further, repeated use of FES is postulated to induce neuroplastic changes such as hippocampal neurogenesis and synaptic plasticity via its therapeutic neuroprosthetic effect through both peripheral (exerkine releases in stimulated muscles [72,73]) and central pathways (increases in peripherally circulating brain-derived neurotropic factor) [73]. FES could also increase neuronal activity (H+ ions) and cerebral blood perfusion in sensorimotor cortices, thalamus, and cerebellum [74,75,76,77], which could also promote neuroplastic changes. Specific to gait and balance control, FES could promote improvements in spatiotemporal gait parameters such as gait speed, cadence, and symmetry [78] and in outcomes of balance control including volitional balance ability assessed via Berg Balance Scale and Timed-up and Go test [79], and possibly reactive balance ability assessed via perturbation-based assessment [80]. Therefore, the addition of FES to unsupervised home-based multicomponent exercise training might provide a safe intervention for OAwS with high motor impairment and potentially result in high adherence.

Thus, the primary purpose of this study was to examine the feasibility of 6 weeks of Smartphone Exercise Training after Stroke (SETS) integrated with FES delivered to lower limb muscles among a single group of OAwS with moderate-to-high motor impairment. Second, we aimed to examine the initial effect of SETS with FES on clinical measures of gait and balance function. We hypothesized that home-based SETS would be feasible (safe, higher adherence, acceptable, user friendly) and yield significant improvements in gait and balance performance pre-to-post training among OAwS, which would exceed the Minimal Clinically Important Difference (MCID).

## 2. Materials and Methods

In this single pre-post study, 14 individuals with chronic stroke who previously completed at least 6 weeks of outpatient PT were included.

### 2.1. Participants

We enrolled 14 adults with chronic hemiparetic stroke (onset >6 months). Participants were recruited from the Greater Chicago Area via flyers, advertisements, and direct physician referrals. All interested participants were first screened on telephone and then screened in person. To pass the telephone screening, participants must be between 60 and 90 years of age, have experienced a stroke more than 6 months prior, able to walk independently with or without an assistive device for at least 300 feet, use a smartphone regularly, and communicate in English. Participants were excluded if they had any neurological condition other than stroke, cardiopulmonary, musculoskeletal, or systemic diagnosis; major surgery; or hospitalization within the last 6 months. Participants who passed the telephone screening were scheduled for an in-person screening. During the in-person screening, participants were excluded if their body weight exceeded 250 lbs, if they had the presence of cognitive impairments (Mini-Mental State Exam score < 25), speech impairment (score of ≤71/100 on Mississippi Aphasia Screening Test), low bone density (T score < −2 on heel ultrasound), or loss of protective sensations (on 5.07/10 g monofilament test). We also tested for possible cardiovascular risk and excluded participants with a resting heart rate > 85% of age-predicted maximal heart rate, resting systolic blood pressure > 165 or diastolic blood pressure > 110 mmHg, shortness of breath, or uncontrolled pain (3/10) to ensure safety during exercise training. Participants were excluded if they had Botox treatment specific to lower limb musculature in the past 3 months or had skin conditions not tolerant to FES therapy. Participants were also excluded if they had a history or current uncontrolled/controlled epilepsy or any other types of seizure disorders, moderate-to-severe spasticity (Ashworth scale > 2), uncontrolled/untreated hypertension, or diabetes. Based on the eligibility criteria, two participants were excluded. One participant had a T score of <−2 (indicating low bone density), whereas the other participant scored < 25 on the Mini-Mental State Examination (indicating cognitive impairment). These exclusions were necessary to ensure participant safety during the study procedures.

### 2.2. Ethical Considerations

This study was approved by the university’s Institutional Review Board (2022-0524) and the clinical trials registry (NCT05849532), and written informed consent was obtained at participant enrollment. Demographics and clinical measures are presented in Table 1.

### 2.3. Intervention

All participants underwent 6 weeks (1 h/day, 3 times/week) of progressive multicomponent exercise training. For the first two weeks, participants underwent supervised onboarding sessions in the lab (i.e., 6 sessions), which included instructions on technology use, safety precautions, and instructions required for achieving training independence at home. After the first two weeks, participants were provided with the equipment package and asked to perform exercises three times per week.

Training components: The intervention was structured around three constructs: dynamic balance, functional strength, and walking (Figure 1), followed by stretching. Each 1 h session included 10 min for each training component and adequate rest breaks. Dynamic balance exercises targeted intentional balance, beginning with mini-lunges, mini-squats, and standing with feet together and progressing to forward lunges and skater lunges. Functional strength exercises target multi-joint muscles required for daily living activities, beginning from sit-to-stand to standing weight shifts and progressing to a stool touch or alternately stepping on a stool. The gait component began from walking on a level ground and progressed to walking on ramps, heel–toe walking, backward walking, and climbing stairs. Lastly, active stretching exercises, including trunk, upper limb, and lower limb stretches progressed in terms of dosage (repetition and holds). The training was delivered via a commercially available software application (TRAINFES Rehab, Version 1.3.0). 

Application setup and use: Sessions were prescribed through a cloud-based dedicated platform (Trainfes Cloud Platform), integrated with the Functional Electrical Stimulation (FES) system for home-based training. The FES system, called Trainfes Advance (Train Health, Palo Alto, CA), was directly connected with the cloud platform and a mobile application, offering a method of training management for both the rehabilitation team and the patients and caregivers. Every patient had an electronic medical and rehabilitation record on the platform. Participants logged onto the application installed on their smartphone and began their assigned training sessions. Each session began with safety precautions, contraindications, and an overview of exercise components. Next, participants donned FES equipment by mimicking the videos in the application. Participants also received electrical stimulation to paretic lower limb muscles during each exercise including the paretic quadriceps, gluteus medius, and tibialis anterior groups based on previous evidence [81,82,83]. The stimulation was delivered using a wireless system with self-adhesive electrodes (size: 5 × 5 cm, type: carbon-rubber, vendor: Axelgaard Manufacturing Co., Fallbrook, CA, USA) and a compensated symmetric biphasic quadratic waveform at a frequency of 25–45 Hz or pulses per second (pps) and pulse width between 150 and 300 μs [81,83]. The stimulation intensity (mA) was tailored to each participant’s functional threshold (i.e., strong but not painful) and was determined during the onboarding sessions. When transitioning to independent home sessions, participants were instructed to dynamically adjust based on the stimulation accommodation. Specifically, participants were instructed to increase the stimulation if they “felt that the intensity was not strong enough” but not exceed the upper limit of 100 mA. Based on the data collected, participants began with a stimulation intensity of 58 ± 11 mA during the first training session and ended with a stimulation intensity of 85 ± 13 mA. The stimulation was triggered by the participants’ movements at the start of each exercise and synchronized with the movement phase using an Inertial Measuring Unit connected via Bluetooth with the FES device and the smartphone app [84,85]. The stimulation lasted the entirety of each exercise, with different muscles stimulated based on the phase of the movement or exercise. For example, during the sit-to-stand movement or exercise, the tibialis anterior was stimulated during the initial preparation phase and the quadriceps was stimulated during the momentum transfer and extension phase.

Participant safety: Before starting the exercises, the software application provided detailed safety precautions to be followed before and during each session. Specifically, participants were advised to inform the caregiver or family member before starting the session; however, they were not needed to have the caregiver present in the same room during exercising. Participants were also advised to perform exercises in a well-lit room with a comfortable temperature and clear of any clutter. Participants were instructed to wear comfortable clothing, wear glasses (if they typically do), and turn off any distractions (TV, radio). During the exercise sessions, participants were asked to stay hydrated and take frequent breaks to avoid exercise-related fatigue. Participants were instructed to perform all balance and strength exercises next to a wall or close to a bed/soft surface while standing on a hard and dry surface free of rugs. Participants were asked to keep a chair in their vicinity while performing exercises or use their assistive device for support. During challenging exercises like climbing stairs, stepping on a stool, or heel-toe walking, participants were asked to have a family member or caregiver present for safety.

### 2.4. Outcome Measures

Our primary measure was feasibility (assessed by safety and adherence) and the secondary measures were clinical measures of participant motivation, acceptability and attitude, usability, and measures of gait and balance function.

Feasibility (safety and adherence): We documented any adverse events during the experimental session. A serious adverse event was an anticipated or unanticipated physical or psychological occurrence resulting in a life-threatening injury or disease, inpatient hospitalization, outpatient admission, persistent or significant disability, or death. A non-serious event was an anticipated or unanticipated physical or psychological occurrence temporarily resulting in injury or disease that did not require medical attention (e.g., muscle sprain, joint stiffness, anxiety, tachycardia). Adherence was the percentage of session completion recorded on the software application and confirmed with the participant.

Motivation: We used the Intrinsic Motivation Inventory (IMI) to assess participant motivation. The IMI is a multidimensional measurement tool used to assess participants’ subjective experiences related to a particular activity in experiments [86]. The IMI includes several subscales, such as Interest/Enjoyment, Perceived Competence, Effort/Importance, Pressure/Tension, Perceived Choice, and Value/Usefulness. Each item within these subscales is rated on a Likert scale, commonly ranging from 1 to 7, where 1 indicates strong disagreement (“not at all true”), 4 indicates being neutral (“somewhat true”), and 7 indicates strong agreement (“very true”). For negative statements within the components, e.g., “I did this activity because I had no choice”, the scoring is reversed. Motivation was assessed during the first and last weeks of training.

Acceptability and attitude: We used a 10-point questionnaire adopted from a previous study to assess the acceptability and attitude toward the exercise intervention [38]. On a scale of 1–3 (1-Agree, 2-Neutral, 3-Disagree), participants were asked questions on whether the SETS intervention helped them with reducing the cost, reducing time and efforts to visit the therapeutic center, whether it was beneficial in improving overall physical, mental, and social health and whether it helped in reducing dependence on a therapist or caregiver. The acceptability and attitude questionnaire was assessed during the first and last weeks of training.

System usability: We used the System Usability Scale, a 10-item questionnaire, to assess participants’ usability of the smartphone application [87]. The participants are asked to score each question on a scale from 1 to 5, ranging from Strongly Disagree (1) to Strongly Agree (5). The scoring is different for odd- and even-numbered items, and the possible score range is 0–100 (https://measuringu.com/sus/, accessed on 1 August 2023). Higher scores indicate better usability of the application. The system usability scale was assessed during the first and last weeks of training.

All outcomes for gait and balance function were based on the global standardized measures of balance and mobility post-stroke [88] and were assessed during the pre- and post-testing sessions. These included the following:

Timed-up and Go test (TUG): The TUG requires a participant to stand up from a chair, walk 3 m, turn around, walk back, and sit “as quickly as possible”, and the time is noted. Higher values indicate poorer performance. For people with chronic stroke, a cutoff score of 13.49 is suggestive of discriminating them with age-matched healthy adults [89]. The MCID considered today is approximately 3 s [90].

Berg Balance Scale (BBS): The BBS assesses static and dynamic balance required during different activities. The BBS is a commonly used scale comprising 14 tasks (4 points each, total of 56 points) among people with chronic stroke. Higher values indicate better performance. An increase in BBS scores by 5 points in assisted walking groups and 4 points in unassisted walking groups is considered as the MCID after exercise training [91].

Mini Balance Evaluation Systems Test (Mini-BESTest): This test assesses four components, including anticipatory balance, reactive balance, sensory orientation, and dynamic gait stability through its 14 task-oriented items with a total scoring of 28 points [92]. Higher values indicate better performance. The Mini-BESTest is valid, has strong inter- and intra-rater reliability, and is commonly used among people with chronic stroke. The MCID was determined to be 3.8 points after rehabilitation among people with stroke in the subacute stage [93].

Ten-Meter Walk test (10MWT): The 10MWT requires a participant to walk at a comfortable walking speed for 10 m and is a commonly used test to assess gait speed. Post-training, in the acute-phase stroke, an improvement of 0.16 m/s was considered as the MCID [94].

The 30 s chair stand test (30CST): The 30CST is a valid test to assess lower limb functional strength [95]. Participants were asked to cross their arms over their chest and instructed to stand up and sit back down in a chair as many times as possible within 30 s, and the number of stand-ups was recorded. An improvement of 2 or more repetitions was considered as the MCID [96].

Short Physical Performance Battery (SPPB): The SPPB is a functional test that measures gait speed, standing balance, and lower extremity strength and endurance. Each test was scored on a scale of 0 to 4 points, with a summary performance score range of 0 to 12 points. Among OAwS, a change of 1 point in the SPPB score was considered the MCID [97].

### 2.5. Sample Size Estimation and Statistical Analysis

We estimated the sample size based on changes in clinical measures of functional mobility (TUG), gait speed (10MWT), dynamic balance (BBS and Mini-BESTest), and overall physical function (SPPB) obtained from the pilot sample (n = 5). The pilot data pre-to-post training results showed a moderate effect size (Cohen’s d: 0.58–0.72), which yielded a sample size of 12 to achieve a statistical power over 80%. Hence, the target sample size for this study was 12 participants. Also, based on the previous literature, a sample size of at least 12 participants is recommended for pilot studies [98].

For statistical analyses, the dataset was first assessed for normality using the Shapiro–Wilk Test. Based on the Shapiro–Wilk test results, i.e., if *p* > 0.05, paired t-tests would be used for parametric variables, and if *p* < 0.05, then the Wilcoxon signed-rank test would be used for non-parametric data to assess the pre-post changes in all outcomes. In addition, for ordinal data like Motivation, Acceptability, and Attitude outcomes, the Wilcoxon signed-rank test would be used to assess pre-post differences. All statistical analyses and data visualization were performed in SPSS version 25 with an alpha level of 0.05.

## 3. Results

Feasibility (safety and adherence): No serious or non-serious adverse events were reported. Frequency statistics showed moderate-to-high adherence (84.17 ± 11.24%) (Table 2). The assessment of clinical measures showed a significant improvement in the system usability scores between S1 and S20 (t = −8.56, *p* < 0.001). Paired t-tests for components of the IMI scale showed significant improvements in interest/enjoyment, perceived competence, effort/importance, value/usefulness, and relatedness (*p* < 0.05 for all). Also, there was an overall improvement in the total scores of acceptability and attitude scores (*p* < 0.05). The domain-wise comparison results for the acceptability and attitude survey are listed in Table 3.

The Shapiro–Wilk test showed a normal distribution for all measures, except the SPPB (*p* > 0.05). For all outcomes of gait and balance function, we observed significant pre-to-post improvements (*p* < 0.05, Figure 2). The results showed pre-to-post improvements in functional mobility (Timed-up and Go test: −6.01 ± 2.53 s [t (11) = 8.21, *p* < 0.001]), dynamic balance (Berg Balance Scale: 5.33 ± 1.87 points [t (11) = −9.85, *p* < 0.001] and the Mini-BESTest: 3.58 ± 1.38 points [t (11) = −9.00, *p* < 0.001]), functional strength (the 30 s chair stand test: 2.58 ± 0.9 repetitions [t (11) = −9.94, *p* < 0.001]), physical function (SPPB: 1.58 ± 0.79 points [t (11) = −6.92, *p* < 0.001]), and overground gait speed (10MWT: 0.18 ± 0.08 m/s [t (11) = −8.095, *p* < 0.001]). Lastly, the pre-post differences in all quantitative measures exceeded the pre-established MCID, except for the Mini-BESTest.

## 4. Discussion

We evaluated the feasibility of 6 weeks of SETS integrated with FES among a single group of OAwS with moderate-to-high motor impairment with results suggesting that the intervention is feasible, i.e., can be safely and consistently practiced in a home environment. Further, SETS integrated with FES also showed up to a 50% improvement in specific outcomes of motivation, acceptability, and attitude toward the intervention and technology system usability from the first to the last week of training. Lastly, SETS integrated with FES exhibited a significant improvement in the measures of gait and balance function pre-to-post training, with differences in most measures exceeding the established MCIDs except for the Mini-BESTest.

Firstly, SETS integrated with FES targeted a marginalized stroke population, i.e., OAwS with moderate-to-high motor impairment and showed feasibility, as shown by the absence of serious or non-serious adverse events and a high adherence rate (average: 84.17%), suggesting that all participants were able to complete the intervention safely and independently. Only two participants showed moderate adherence as compared to others, one of them had a history of back pain, and they reported back pain (visual analog scale—VAS:2/10) during the home training sessions, which could potentially have resulted in the 60% adherence. The other participant showed 70% adherence to the intervention with no complains of pain; however, this participant was the oldest of the cohort (73.8 years) and had the highest motor impairment (CMSA leg: 2/7), which could have been the factors affecting their intervention adherence.

Our feasibility findings (safety and adherence) align with some previous evidence from efficacious interventions delivering home-based exercises among people with stroke [42,99,100]. However, most of these interventions were supervised, either delivered by a healthcare professional (e.g., PT, health coaches) [29,30,31,32] or by caregivers [33,34], whereas the current intervention was an independent intervention delivered via a smartphone. The feasibility seen with SETS integrated with FES could have resulted from the possible twofold benefits achieved from principles of action of digital technology and FES. Delivering the intervention via digital technology could have assisted in maintaining participant engagement and motivation to complete the exercise program in an unsupervised setting. Typically, conventional home exercise programs prescribed after outpatient physical therapy discharge include general exercises delivered via paper booklets or brochures [40,101,102], which might not be engaging and, thus, account for low adherence to the training programs. SETS, on the other hand, was delivered via digital technology and consisted of features such as video-based guidance, frequent positive reinforcements, self-evaluation of balance confidence, and exertion. Collectively, these features could have served to ensure participant engagement during training sessions [37,103,104,105], thus promoting adherence to the program as seen in the study results. Secondly, the integration of FES into SETS could possibly have ensured safety and promoted the completion of exercises via its peripheral mechanisms of action. Specifically, applying FES to paretic lower limb muscles during challenging gait and balance exercises could have assisted in the instant activation of neuromuscular units (e.g., muscle spindles, neuromuscular junctions) and skin afferents (e.g., somatosensory cutaneous receptors) [106] to facilitate the safe completion of exercises that, otherwise, would have been too difficult for OAwS with high motor impairment. With such immediate external assistance provided during challenging exercises, FES could have also ensured adherence to the program. In addition, the application of FES during gait and balance exercises might have assisted participants in completing exercises with less energy expenditure [107,108] or reduced perceived workload [109,110]. Other elements of the SETS integrated with FES that could have promoted improved adherence to the program might be the structured onboarding sessions, safety protocols, and regular text reminders to complete the training sessions [111].

Participants also reported significant improvements in system usability between the first and last weeks of training, suggesting increased comfort with technology use. Further, participants reported an overall pre-to-post enhancement in outcomes of motivation, acceptability, and attitude. Specifically, better scores were reported on domains of interest, perceived competence, effort, value or usefulness, and relatedness. These results were in line with previous studies have shown that increased enjoyment [112] and increased effort [113,114] could be associated with improved motor performance. Our results showed no pre-to-post training differences in domains of pressure/tension and perceived choice, which were in-line with previous studies that showed no differences in domains of pressure/tension with shorter training durations [115]. In this study, the participants could have perceived no differences in these domains as the training was limited to 6 weeks. Alternatively, no changes in these domains might be related to the ceiling effects of the outcomes, as all the participants demonstrated higher baseline scores (>5.25, 4 = somewhat true). Hence, differences in these domains might not be the most appropriate elements associated with training-induced changes. Other elements that showed improvement were acceptability and attitude, with bigger changes seen on the items of “reducing efforts to visit the therapeutic center”, “reduced dependence on the therapist for exercises”, and “reduced dependence on caregiver”, which could explain the higher usability satisfaction and higher motivation pre-to-post training (Table 3). Such a relationship between patient-centered factors of motivation, effort, and acceptability/usage was previously shown to influence adherence to training among people with stroke [116,117,118].

In addition to showing feasibility and improvements in the outcomes of motivation, acceptability and attitude, and usability, significant improvements in the clinical outcomes of functional mobility, gait, dynamic balance and functional strength were observed, all of which were components of SETS exercises. These results were in line with previous home-based exercise programs that have shown improvements in training-specific components (e.g., Wolf-motor function, physical activity assessed via number of steps). However, previous studies did not specify whether the training-induced differences exceeded the established MCIDs [119,120]. In the current study, pre-post improvements in all clinical outcomes exceeded the established MCID, except for the Mini-BESTest. Pre-post training performance on the Mini-BESTest did not exceed the established MCID, which could possibly be because the SETS with FES intervention did not include components to train reactive balance and vestibular system. The Mini-BESTest examines multiple components of balance control, including sensory orientation, dynamic and reactive balance, vestibular and non-vestibular components, and gait-related functional mobility [121,122]. Although SETS with FES included exercises targeting most of these components, reactive and vestibular balance components were not included. Future studies could incorporate these components into their intervention to achieve more clinically meaningful improvements in assessments such as the Mini-BESTest. Overall, the improvements in clinical outcomes observed with 6 weeks of SETS with FES could possibly have resulted from the potential neurofacilitatory effect of FES [123,124]. Specifically, the addition of FES to the SETS could possibly have aided in completing the prescribed exercises by activating the paretic muscles, thus enhancing voluntary movement control, providing exercise practice, and improving functional performance pre-to-post training. However, since this study did not incorporate true control group(s), i.e., SETS without FES and traditional home exercise programs (no SETS and/or no FES), the current study is limited in its ability to identify the differential effects of smartphone-based digital technology and Functional Electrical Stimulation.

Smartphone-based exercise training (SETS) could offer some advantages over traditional supervised or unsupervised training. First, SETS provides real-time feedback and guidance through video demonstrations, which could enhance participant engagement and motivation. Unlike supervised training that requires the presence of a therapist, SETS allows for greater participant flexibility and independence, making it more accessible for individuals with scheduling constraints. Additionally, SETS incorporates features such as the self-evaluation of balance confidence and exertion, which might not be available in unsupervised paper-based programs. These features could contribute to the high adherence rates and improvements in outcomes as seen in the study results.

### Limitations and Future Directions

One study limitation could be the sample size. However, the current sample was derived from a priori power analysis based on a preliminary sample (n = 5), which is also a recommended sample size for pilot studies [98]. Future studies can include larger, heterogenous samples to validate our results. Second, a limitation of this study was the lack of a control group, which could significantly limit the ability to compare the effectiveness of this intervention with other existing ones and decipher the differential effects of smartphone-based technology and FES on study outcomes. Future studies might consider conducting randomized controlled trials with multiple comparison groups including SETS without FES and traditional home exercise programs to validate the efficacy of SETS integrated with FES. Although this study did not include electromyography (EMG) or biomechanical assessments to evaluate the specific effects of FES on muscle activation and movement patterns, the primary focus was on assessing the feasibility and clinical outcomes of the SETS intervention. The observed improvements in gait speed, balance, and functional mobility provide preliminary evidence of the intervention’s potential benefits. Future studies could incorporate bioelectrical and/or biomechanical assessments to explore the underlying mechanisms of FES and its contribution to motor recovery in greater detail. Further, the SETS with FES did not incorporate real-time performance feedback, future studies might consider adaptive algorithms for incorporating factors like feedback, including knowledge of performance and results, choices for training progression, all of which can further personalize and optimize training protocols based on individual needs.

## 5. Conclusions

Smartphone-based Exercise Training after Stroke (SETS), an intervention that integrates technology and Functional Electrical Stimulation (FES) to deliver task-specific exercises, is a feasible paradigm for home-based training among people with chronic stroke, especially for those with moderate-to-high motor impairment and older age (>55 years). Smartphone technology-delivered exercise training can potentially assist in bridging the gap for those unable to access traditional, clinic-based PT due to mobility, economic, or logistic constraints.

## Figures and Tables

**Figure 1 sensors-25-01254-f001:**
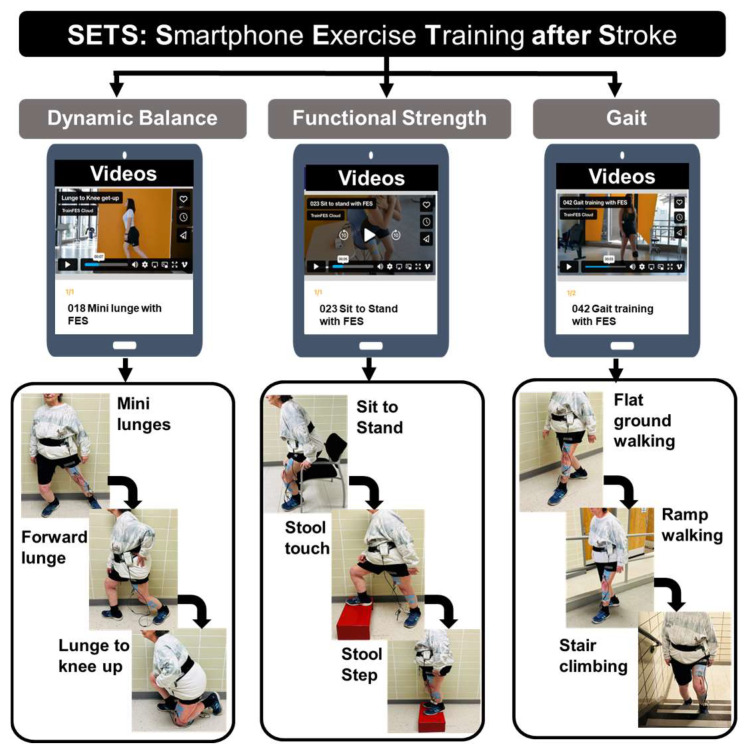
Intervention protocol for SETS (Smartphone Exercise Training after Stroke) with three domains, including dynamic balance, functional strength, and gait. Each component included three levels of exercise progression, examples of which are provided under each domain.

**Figure 2 sensors-25-01254-f002:**
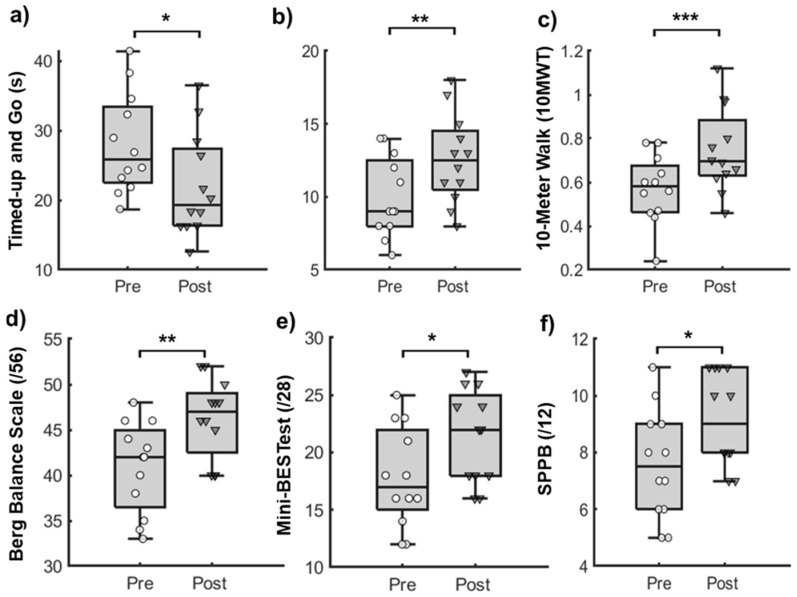
Box and whisker plots with individual data points for outcomes of gait and balance function. Boxes represent the interquartile ranges (25–75%) of data variance, bold lines within boxes represent the mean, and whiskers represent the standard deviations for (**a**) Timed-Up and Go test (TUG), measured in seconds; (**b**) 30-Second Chair Stand Test (30CST), measured in repetitions; (**c**) 10-Meter Walk Test (10MWT), measured in meters per second; (**d**) Berg Balance Scale (BBS), scored out of 56; (**e**) Mini-Balance Evaluation System Test (Mini-BESTest), scored out of 28; and (**f**) Short Physical Performance Battery (SPPB), scored out of 12 assessed during pre- and post-testing sessions. * *p* < 0.05, ** *p* < 0.01, *** *p* < 0.0001.

**Table 1 sensors-25-01254-t001:** Demographics and baseline clinical characteristics of research participants with their respective means and standard deviations.

Variables	n = 12 Mean (SD)
Age, y	67.58 (5.36)
Sex, M/F	6/6
Race	African Americans (12)
Height, m	1.69 (0.10)
Weight, kg	80.75 (16.69)
BMI, kg/m^2^	28.74 (4.74)
Hemi-side, R/L	5/6
Chronicity, y	9.16 (2.96)
Type of stroke, H/I	7/5
AFO/No AFO	9/3
CMSA (leg) (out of 7)	2.41 (0.64)
Fugl Meyer (LE) (out of 28)	18.41 (3.84)
MMSE	26.17 (1.14)
MoCA	20.17 (1.67)

Abbreviations: y: Years; M/F: Male, Female; m: Meter; kg: Kilogram; kg/m^2^: Kilogram per meter square; R/L: Right/Left; H/I: Hemorrhagic/Ischemic; AFO/No AFO: Ankle-foot orthosis; CMSA: Chedoke McMaster Assessment; MMSE: Mini-Mental State Examination; MoCA: Montreal Cognitive Assessment Scale.

**Table 2 sensors-25-01254-t002:** Feasibility outcomes (safety and adherence) during the Smartphone Exercise Training after Stroke (SETS) intervention.

Participant	Safety (Adverse/Non-Adverse Events and Other Symptoms)	Adherence (% Completion)
1	None	85
2	None	70
3	None	90
4	None	90
5	Back pain (VAS: 2/10)	60
6	None	95
7	Fatigue (VAS: 2/10)	80
8	Knee pain (VAS: 3/10)	80
9	Fatigue (VAS: 3/10)	85
10	None	95
11	Fatigue (VAS:2/10)	80
12	None	100

Abbreviations: %: Percentage, VAS: Visual Analog Scale.

**Table 3 sensors-25-01254-t003:** Means and standard deviations in measures of motivation, acceptability and attitude, and system usability were assessed after the first and last training sessions.

Outcome	Pre (Week 1) Median [IQR]	Post (Week 6) Median [IQR]	*p* Value
Interest/Enjoyment	5 [4.25–5]	6 [5.25–7]	0.006 *
Perceived competence	4.5 [4–5.75]	6 [6–7]	0.011 *
Effort/Importance	5 [4–5]	6 [5–7]	0.02 *
Pressure/Tension (score reversed)	5.5 [4–6.75]	5 [4–6.75]	0.58
Perceived choice	5.5 [5–6.75]	6 [4.25–6.75]	0.67
Value/Usefulness	5 [4–5.75]	6 [5.25–7]	0.013 *
Relatedness	6 [5–6]	6 [6–7]	0.015 *
**Acceptability and Attitude** **(Questionnaire)**	**Pre (S1)**	**Post (S20)**	
Reduce costs	0/33.33	25/16.67	
Reduce time	0/50	25/8.33	
Help perform exercises decently	0/58.33	66.67/0	
Reduce dependence on the therapist for exercises	0/50	83.33/0	
Reduce dependence on the caregiver	0/50	83.33/0	
Recommend to other friends	0/83.33	50/0	
Continue to use application after termination	0/25	75/0	
Continue to use device after termination	0/83.33	50/0	
Helped improve physical health	0/16.67	83.33/0	
Helped improve mental health	0/25	50/0	
Helped improve social health	0/83.33	16.67/1.83	
**Total points (Median [IQR])**	32 [31.25–33]	19 [18–20]	0.002 *
**System Usability Scale (Mean ± SD)**	61.75 ± 10.91	75.00 ± 11.74	<0.001 *

Abbreviations: IQR: Interquartile Ranges, *: *p* < 0.05.

## Data Availability

Data will be made available upon request.

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
