# Peer review of "Feasibility of Smartphone-Based Exercise Training Integrated with Functional Electrical Stimulation After Stroke (SETS): A Preliminary Study"

_sensors, 2025, doi:10.3390/s25041254_

Round 1
Reviewer 1 Report
Comments and Suggestions for Authors
This study explored the safety of combining smartphone-based post-stroke exercise training with functional electrical stimulation for improving the motor function of elderly stroke patients. The study recruited 12 elderly patients who had suffered from stroke for more than 6 months and received 6 weeks of training in gait, functional strength, and dynamic balance. They were tested before and after the training with a 10-meter walking test and Mini-BESTest, and their performance before and after the training was compared. The results of the study showed that the safety and compliance of the combined treatment were good, and the gait performance and balance function of the patients were improved. Some issues should be addressed:
1. The enrolled patients in this study were 14. Why two patients were excluded?
2. The comparison of count/ordinal datasets, such as Motivation, Acceptability & Attitude, should use chi-square test instead of paired t-test.
3. What role does smartphone-based exercise training play in this study? What are the special properties of smartphone-based exercise training that enable combined therapy to produce better results compared to supervised or unsupervised training therapy? In other words, if a supervisor provides patients with information similar to that of a smartphone during training, can it also produce the same effect? The author should conduct in-depth discussions on this.
Author Response
We thank the reviewer for their time, comments and feedback.
Please see the point-by-point responses to the reviewer comments in the attached word document.

Reviewer 2 Report
Comments and Suggestions for Authors
This manuscript investigates the feasibility of a home-based rehabilitation program integrating Smartphone-based Exercise Training after Stroke (SETS) with Functional Electrical Stimulation (FES) in individuals with moderate-to-high motor impairment post-stroke. The study effectively highlights the potential of digital health interventions in enhancing rehabilitation accessibility for older adults with stroke (OAwS). The results suggest promising feasibility, high adherence rates, and functional improvements in gait and balance measures. The methodology is well-structured, and the study addresses an important clinical need for home-based stroke rehabilitation.
However, several limitations and areas for improvement need to be addressed:
- The Introduction section briefly mentions FES neurofacilitation, but the manuscript lacks an in-depth discussion of how FES enhances gait and balance control. The neuromechanical effects of FES, such as activation of muscle spindles, proprioceptive enhancement, and sensory feedback modulation, should be elaborated. The potential for cortical plasticity via FES-driven neuromodulation is a crucial aspect that remains unexplored.
- Incomplete FES Parameter Details (Page 5, Lines 202-206). While the study appropriately applies FES to the paretic quadriceps, gluteus medius, and tibialis anterior muscles, several technical details are missing:(a) Stimulation intensity range (45-100mA) is quite broad. Did each participant receive the same threshold intensity, or was the intensity dynamically adjusted based on fatigue or progress?(b) Stimulation timing and triggering mechanism need clarification. Was the stimulation initiated manually, based on EMG activity, or synchronized with gait phase sensors?(c) Electrode specifications are absent. The manuscript should specify the size, type, and vendor of the electrodes used.
- The absence of a control group (e.g., SETS without FES, traditional home therapy) limits the ability to isolate the effects of FES from the smartphone-based training component. While the results indicate significant functional improvements, it remains unclear whether these gains were primarily driven by FES, the structured exercise regimen, or the engagement features of digital technology. Even within a pilot study, incorporating historical data comparisons (e.g., expected natural recovery trajectories without FES) could enhance the study's scientific rigor.
- No electromyography (EMG) or biomechanical assessments were conducted to evaluate how FES influenced muscle activation or movement kinematics. While functional improvements in gait and balance were observed, the specific contribution of FES to muscle activation patterns remains unclear.
Author Response
We thank the reviewer for their time, comments and feedback.
Please see the point-by-point response in the attached word document.

Round 2
Reviewer 2 Report
Comments and Suggestions for Authors
The authors have addressed all my comments, and I have no further suggestions.